# Prophylactic Antibiofilm Activity of Antibiotic-Loaded Bone Cements against Gram-Negative Bacteria

**DOI:** 10.3390/antibiotics11020137

**Published:** 2022-01-21

**Authors:** Andréa Cara, Tristan Ferry, Frédéric Laurent, Jérôme Josse

**Affiliations:** 1Centre International de Recherche en Infectiologie, CIRI, Inserm U1111, CNRS UMR5308, ENS de Lyon, UCBL1, 69007 Lyon, France; andrea.cara@inserm.fr (A.C.); tristan.ferry@univ-lyon1.fr (T.F.); frederic.laurent@univ-lyon1.fr (F.L.); 2Université Claude Bernard Lyon 1, 69100 Lyon, France; 3Service de Maladies Infectieuses, Hôpital de la Croix-Rousse, Hospices Civils de Lyon, 69002 Lyon, France; 4Centre Interrégional de Référence des Infections Ostéo-Articulaires Complexes (CRIOAc Lyon), Hospices Civils de Lyon, 69002 Lyon, France; 5Laboratoire de Bactériologie, Institut des Agents Infectieux, Hôpital de la Croix-Rousse, Hospices Civils de Lyon, 69002 Lyon, France

**Keywords:** antibiotic-loaded bone cement, prosthetic joint infection, biofilm, *Pseudomonas aeruginosa*, *Escherichia coli*, *Klebsiella pneumoniae*

## Abstract

Gram-negative bacilli can be responsible for prosthetic joint infection (PJI) even if staphylococci are the main involved pathogens. Gram-negative PJIs (GN-PJI) are considered difficult-to-treat infections due to the increase in antimicrobial resistance and biofilm formation. To minimize the risk of infection in cases of arthroplasties with cemented prosthesis, bone cement can be loaded with antibiotics, especially gentamicin. In this study, we aimed to compare the prophylactic antibiofilm activity of ready-to-use antibiotic-loaded bone cements (ALBC), already commercialized or new prototypes. We compared ALBCs containing gentamicin alone, gentamicin plus vancomycin, gentamicin plus clindamycin, gentamicin plus Fosfomycin, and fosfomycin alone, to plain cement (no antibiotic); these comparisons were conducted to investigate the biofilm formation of three strains of *Escherichia coli*, three strains of *Pseudomonas aeruginosa* and two strains of *Klebsiella pneumoniae*, with or without specific resistance to gentamicin or fosfomycin. We reported that ALBC containing gentamicin and clindamycin (COPAL G+C) seems to be the most interesting ALBC of our tested panel for the prevention of biofilm formation by gentamicin-susceptible strains, even if clindamycin is not effective against Gram-negative bacteria. However, gentamicin-resistant strains are still a problem, and further studies are needed to identify an antibiotic to associate with gentamicin for an efficient dual ALBC against Gram-negative bacteria.

## 1. Introduction

Prosthetic joint infection (PJI) is one of the most serious complications of total hip or knee joint arthroplasties and requires a multidisciplinary approach for successful management [1,2]. The most common infective agents in PJIs are Gram-positive bacteria, especially *Staphylococcus aureus* and *Staphylococcus epidermidis* [3,4]. However, Gram-negative bacilli can also be responsible for PJIs. Gram-negative PJIs (GN-PJIs) represent approximately 12–15% of total PJIs [5,6] and can occur through contamination during surgery or through hematogenous seeding [5]. Even if they represent a small volume of total PJIs, GN-PJIs are considered more difficult to treat, notably due to their increasing antimicrobial resistance [6,7]. Monomicrobial GN-PJIs mostly involve *Pseudomonas aeruginosa*, *Escherichia coli* and *Klebsiella pneumoniae* [5,6]. These 3 species can form a biofilm on prosthesis materials, contributing to difficulties in the treatment of GN-PJIs [8,9]. Biofilm corresponds to a structured community of bacteria embedded in a self-produced extracellular matrix. This bacterial organization provides to bacteria the capacity to tolerate antibiotic treatments at clinical concentrations [10].

To minimize the risk of infection in cases of arthroplasties with cemented prosthesis, the bone cement can be loaded with antibiotics [11]. The objective of this practice is to prevent the development of an infection in the joint at the local level and, notably, to avoid the formation of a biofilm on the prosthesis. Bone cements are composed of polymethyl methacrylate (PMMA). They are mostly used in orthopedic surgery for prosthesis fixation, and as spacer in cases of 2-step exchange in the case of PJI. In prophylactic situations and in cases of cemented prosthesis implantation, bone cements can be loaded with low doses of antibiotics (between 0.5 and 2 g of antibiotics for 40 g of PMMA) to prevent PJIs with a limited impact on the mechanical properties of cement [12]. Commercial ready-to-use antibiotic-loaded bone cements (ALBC) mostly include gentamicin with the potential addition of another antibiotic, such as vancomycin or clindamycin [13]. ALBCs can also be handmade, with the addition of antibiotics by the surgeons to classic PMMA bone cements.

The interest in using ALBCs for prosthesis fixation during revision arthroplasties has been highlighted in several studies, even if its use is still debated for primary arthroplasties [14,15,16,17]. In a recent meta-analysis, Sebastian et al. reported that antibiotic-loaded bone cements (ALBCs) are effective in reducing PJI following primary total joint arthroplasty, with a reduction in risk of between 20 and 84% [14]. Similar results were reported in total hip arthroplasty [15]. *S. aureus* and coagulase-negative staphylococci (CoNS) are used as model microorganisms in in vitro studies exploring the potential of ALBCs for the prevention of PJIs [18,19,20]. We previously tested, in vitro, the prophylactic antibiofilm effect of commercial ready-to-use dual ALBCs against clinical strains of *S. aureus* and CoNS [18]. We reported that adding vancomycin or clindamycin to gentamicin is of particular interest, especially when ALBCs were tested against gentamicin-resistant staphylococci. Due to the high rate of gentamicin resistance among CoNS, combining gentamicin with another antibiotic in ALBCs appears relevant to prevent PJIs in cemented prothesis arthroplasties [21].

Few studies have been performed regarding the prophylactic effect of ALBCs against Gram-negative bacteria. In 2013, Chang et al. tested various ALBCs and reported that gentamicin had good antibacterial activity against various bacteria, including *P. aeruginosa* and *E. coli* [22]. However, gentamicin-resistant Gram-negative bacteria cannot be neglected, as they could represent up to 18% of total isolates, depending on the studies [23,24]. Combining another antibiotic with gentamicin in ALBCs to prevent GN-PJIs could be an effective choice. Unfortunately, the proposed additive commercial antibiotics, vancomycin or clindamycin, are not effective against Gram-negative bacteria. However, the presence of vancomycin or clindamycin in addition to gentamicin in ALBCs could impact the release of the antibiotic, leading to a modulation of the prophylactic effect of ALBCs against biofilm formation.

In this study, we aimed to test the prophylactic antibiofilm effect of commercialized ready-to-use ALBCs (loaded with gentamicin alone, gentamicin plus vancomycin or gentamicin plus clindamycin) against Gram-negative bacteria. We also proposed testing the antibiofilm effect of new ALBC prototypes loaded with gentamicin and/or fosfomycin, directly provided as ready-to-use ALBCs by the manufacturers. This antibiotic has a broad antibiotic spectrum and shows an interesting synergism with gentamicin against the biofilm of *E. coli* and *P. aeruginosa* [25].

## 2. Methods

### 2.1. Bacterial Strains

Eight clinical strains of Gram-negative bacteria with specific antibiotic resistances (gentamicin or fosfomycin) were used in this study: 3 strains of *Pseudomonas aeruginosa*, 3 strains of *Escherichia coli* and 2 strains of *Klebsiella pneumoniae*. These clinical strains were isolated during routine work performed at the Bacteriology Department of Hôpital de la Croix-Rousse, Hospices Civils de Lyon, where identification was performed with MALDI-TOF (Vitek MS, Biomérieux, Marcy-l’Étoile, France). Resistance profiles were determined with Vitek 2 (Biomérieux, France). All the strains were tested with the Crystal Violet method beforehand, to ensure that they can form at least a moderate biofilm regarding Stepanovic’s classification [26]. Antibiotic susceptibilities for gentamicin and fosfomycin are presented for each strain in Table 1.

### 2.2. Antibiotic-Loaded Bone Cements

Six different bone cements were used in this study. Four of them were already commercialized by Heraeus Medical: plain cement (PALACOS R, without antibiotic), cement loaded with gentamicin alone (PALACOS R+G, called G thereafter), cement loaded with gentamicin plus vancomycin (COPAL G+V, called G+V thereafter) and cement loaded with gentamicin plus clindamycin (COPAL G+C, called G+C thereafter). Two other cements loaded with gentamicin plus fosfomycin (COPAL G+F, called G+F thereafter) or with fosfomycin (COPAL F, called F thereafter) were specially prepared for this study. Disk-like specimens (diameter 2.5 cm, height 1.0 cm) were used. Specific antibiotic loads for each cement are presented in Table 2.

### 2.3. Preparation of ALBC Elution Solutions

We used the same protocol as in Cara et al. [18]. Briefly, Elution solutions containing antibiotics released from ALBCs were used to evaluate the prophylactic antibiofilm effect of the cements. Disk-like specimens were incubated in 20 mL of Tryptic Soy Broth (TSB Bacto, BD, Le-Pont-de-Claix, France) supplemented with 1% of glucose in a Falcon tube of 25 mL. The ALBCs were incubated for 1 to 9 days at 37 °C. The media were changed daily. For the biofilm formation experiments, ALBCs elution solutions from Day 1, Day 3 and Day 9 were used.

### 2.4. Determination of the Prophylactic Antibiofilm Effect of ALBCs Elution Solutions

Overnight bacterial cultures in liquid Brain–Heart Infusion (BHI, Biomérieux, France) were standardized to OD600 = 1 ± 0.05 before being diluted at 1:100 in ALBCs elution solutions (Day 1, Day 3, and Day 9), and 100 µL was added in a 96-well plate (Greiner Bio-One, Frickenhausen, Germany). Plates were incubated for 24 h at 37 °C in a humid atmosphere. Then, the supernatant was removed, and biofilms were washed for 45 min using BiofilmCare, with a smooth washing method that favors the preservation of the biofilm [27]. Biofilms were then resuspended in 200 µL of Phosphate Buffer Solution (PBS, Gibco, Paisley, UK) by scraping the wells using sterile pipette tips and sonicating for 10 min at 40 Hz (Bactosonic, Bandelin, Berlin, Germany). The number of viable bacteria inside the biofilm was evaluated with plate counting on blood agar plates (COS, Biomérieux, France).

### 2.5. Graphical Representation and Statistical Analysis

Three independent experiments in technical triplicate (3 wells for each condition for each experiment) were performed. Results were presented as the number of viable bacteria inside the biofilm after 24 h of incubation. Data (9 values per condition) were presented as histograms (mean with SD). We compared the data from each day using Kruskal–Wallis tests and Dunn’s multiple comparisons. We performed a first test to compare G, G+V, G+C, G+F and F to the control condition (plain cement). Then, we performed a second test to compare G, G+V, G+C, G+F and F with each other. All analyses were performed using Prism 8 software (GraphPad Software Inc., San Diego, CA, USA).

## 3. Results

### 3.1. Prevention of Biofilm Formation by ALBCs with Gentamicin- and Fosfomycin-Susceptible Strains

We first evaluated the antibiofilm prophylactic effect of ALBCs against multi-susceptible strains of *E. coli*, *K. pneumoniae* and *P. aeruginosa* (Figure 1). For *E. coli*, all ALBCs had a significant prophylactic effect against biofilm formation on Day 1 and Day 3 when compared to plain cement, except for F on Day 3 (Figure 1A). We observed a decrease in biofilm formation of 5-Log at least. At these times, the presence of vancomycin in G+V did not impact its prophylactic activity against biofilm formation when compared to G. Similar observation were made for G+C, where the addition of clindamycin and the highest concentration of gentamicin (G+C contains 1 g of gentamicin instead of 0.5 g for the other gentamicin-containing ALBCs) did not modulate the antibiofilm effect of ALBC when compared to G cement. Regarding G+F, its prophylactic antibiofilm activity was similar to G cement, suggesting that fosfomycin did not provide an additional antibiofilm effect to G+F in comparison to G. On day 9, no ALBC was able to decrease the number of viable bacteria, except for G+C, which kept its antibiofilm prophylactic effect (Figure 1A).

A similar pattern was observed for *K. pneumoniae*. All ALBCs decreased biofilm formation on Day 1 when compared to plain cement, except for cement F (Figure 1B). On Day 3, only G+V, G+C and G+F kept their ability to decrease biofilm formation, with significant differences between G vs. G+V and G vs. G+C (Figure 1B). However, as observed for *E. coli*, only G+C significantly decreased biofilm formation on Day 9 when compared to plain cement (Figure 1B). Concerning *P. aeruginosa*, significant decreases were observed for all ALBCs when compared to plain cement from Day 1 to Day 9, except for the cement F (Figure 1C). On Day 9, the highest *p* value (*p* < 0.0001) was observed for G+C when compared to plain cement (Figure 1C).

Globally, G+V and G+F had similar effects when compared to G. Adding vancomycin did not seem to modify the activity of gentamicin in G+V and adding fosfomycin to gentamicin did not provide additional antibiofilm effect to G+F. The remaining effect of G+C observed on Day 9, compared to other ALBCs, could be explained by the highest concentration of gentamicin in this ALBC.

### 3.2. Prevention of Biofilm Formation by ALBCs with Gentamicin-Resistant Strains

Next, we tested the antibiofilm prophylactic effect of ALBCs against gentamicin-resistant strains of *E. coli*, *K. pneumoniae* and *P. aeruginosa* (Figure 2). For the gentamicin-resistant *E. coli*, only G+C, G+F and F showed a significant decrease in the number of viable bacteria inside the biofilm on Day 1 when compared to plain cement, G and G+V (Figure 2A). In this case, adding fosfomycin to gentamicin provided the best decreasing effect on biofilm formation with the highest *p* values (Figure 2A). In this situation, it seems that fosfomycin has a prophylactic antibiofilm effect.

A similar pattern was observed for the gentamicin-resistant *K. pneumoniae* on Day 1, but in this case, G+C achieved the best antibiofilm prophylactic effect (Figure 1C). However, looking on Day 3 and Day 9, no significant difference was observed for ALBCs when compared to plain cement for these 2 strains, except for a small decrease for G+F and F on Day 3 (Figure 2A,B).

For *P. aeruginosa*, resistance to gentamicin seemed to have less of an impact on the biofilm formation. Indeed, all ALBCs containing gentamicin had similarly decreased biofilm formation, with no differences between them (Figure 2C).

Globally, adding fosfomycin to gentamicin partially rescued the capacity of ALBCs to prevent biofilm formation by gentamicin-resistant strains, but only for the *E. coli* and the *K. pneumoniae* on Day 1. The highest concentration of gentamicin in G+C improved the antibiofilm effect of the ALBCs, only on Day 1, for the gentamicin-resistant *E. coli* and *K. pneumoniae*.

### 3.3. Prevention of Biofilm Formation by ALBCs with Fosfomycin-Resistant Strains

In this part, we only tested fosfomycin-resistant *P. aeruginosa* and *E. coli* as we met some difficulties in finding a fosfomycin-resistant *K. pneumoniae* in our collection of clinical strains. As expected, fosfomycin alone was not efficient against these fosfomycin-resistant strains (Figure 3).

Concerning the fosfomycin-resistant *E. coli*, we observed that all ALBCs containing gentamicin had a strong effect on the decrease in biofilm formation on Day 1 (Figure 3A). On Days 3 and 9, G+F and G+V followed the same pattern as the G cement regarding the decrease in biofilm formation, with an absence of effect on Day 9 (Figure 3A). G+C kept a good prophylactic effect against biofilm formation, potentially due to the highest concentration of gentamicin in this type of ALBC.

Regarding fosfomycin-resistant *P. aerugionosa*, a similar pattern was observed (Figure 3B). All ALBCs containing gentamicin had a strong antibiofilm effect on Day 1, but on Days 3 and 9, G and G+F lost some efficacy, even if a significant difference was observed when compared to plain cement (Figure 3B). For G+V, an interesting effect against biofilm formation persisted until Day 9, raising questions about the role of vancomycin presence in the ALBC on the release of gentamicin, as vancomycin is not active against Gram-negative bacteria. For G+C, this ALBC revealed the best activity against fosfomycin-resistant *P. aeruginosa*.

Gentamicin-containing ALBCs globally kept a good antibiofilm effect against fosfomycin-resistant strains. However, the use of G+C provided higher decreases regarding biofilm formation for the two tested strains.

## 4. Discussion

In this study, we investigated the in vitro prophylactic antibiofilm activity of commercialized ready-to-use ALBCs against several strains of *E. coli*, *K. pneumoniae* and *P. aeruginosa*, which are three species often identified in GN-PJIs [5,6]. We used an in vitro model based on biofilm formation assays in elution solutions from ALBCs [18]. ALBCs were materialized as cylindric specimens made of commercialized PALACOS (plain cement and G) or COPAL (G+V and G+C). However, we also tested new prototypes (not commercialized) of ready-to-use COPAL bone cements loaded with gentamicin and fosfomycin or fosfomycin alone. Our aim was to test them as an alternative to dual the ALBCs, COPAL G+C and G+V, where both antibiotics would be active against Gram-negative bacteria (vancomycin and clindamycin were not active against Gram-negative bacteria).

Globally, we observed that all gentamicin-containing ALBCs have the same behavior: a good, but not strong, activity against gentamicin-susceptible strains (Figure 1 and Figure 3), but an absence of activity against the gentamicin-resistant enterobacteria, especially with elution solutions from Day 3 and Day 9 (Figure 2A,B). However, COPAL G+C revealed a higher antibiofilm effect than the other ALBCs. As clindamycin is not active against Gram-negative bacteria, we strongly suggest that this higher activity is due to the higher quantity of loaded gentamicin. Indeed, the ready-to-use G+C is loaded with 1 g of gentamicin, whereas G, G+V and G+F are loaded with 0.5 g of gentamicin. We suggest the same in our previous study concerning the effect of G+C against staphylococci [18]. Moreover, Karaglani et al. has recently reported that COPAL G+C provides a significantly higher (more than double) gentamicin elution than homemade PALACOS cement with the same concentration of 2.4% gentamicin [28]. Earlier results from Ensing et al. confirmed this synergistic elution booster effect of the gentamicin–clindamycin combination over gentamicin only [29]. Finally, even if clindamycin is not interesting to treat Gram-negative bacteria, the higher concentration and better release pattern of gentamicin in COPAL G+C could be of interest to fight Gram-negative PJI. However, it is difficult to settle the question of whether the improved antibiofilm effect of PALACOS G+C against Gram-negative bacteria is due to the highest concentration of gentamicin, or to the effect of clindamycin presence in the ALBC on the release of gentamicin. One solution would have been to test another ALBC loaded with 1 g of gentamicin, or an ALBC loaded with 0.5 g of gentamicin and 1.5 g of clindamycin. However, these types of ALBCs are not available as ready-to-use ALBCs. Another solution would have been to manually add clindamycin to G cement to evaluate its impact on the antibiofilm effect, in comparison to a classic G cement. However, the manual addition of antibiotic to a ready-to-use ALBC could result in a heterogenous repartition of the added antibiotic in the ALBC, creating another bias. Moreover, manual addition of an antibiotic to cement could alter the mechanical properties of the cement, which could be deleterious for prosthesis fixation.

Regarding fosfomycin, it did not seem interesting in our conditions. Indeed, results for the ALBC containing only fosfomycin were significantly different from those for plain cement, only on Day 1, for the gentamicin-susceptible *E. coli* strain and the gentamicin-resistant *E. coli* strain (Figure 1A and Figure 2A). Regarding G+F that contains gentamicin (0.5 g) and fosfomycin (1.5 g), its efficacy was similar to the ones observed for G and G+V, suggesting that its activity is mostly due to gentamicin. These results are surprising, as fosfomycin was shown to be a good antibiotic to treat biofilm, with a synergistic effect when it is associated with gentamicin [25]. In the context of urinary tract infections, a significant reduction in biofilm was reported for 38 tested clinical strains of *E. coli* [30]. However, in these two studies, fosfomycin was used against formed biofilms and was not investigated regarding its capacity to prevent biofilm formation. Our two hypotheses regarding the lack of effect for fosfomycin-containing ALBCs in our conditions is (i) a too-low quantity of fosfomycin in ALBCs or (ii) an issue regarding its release. Further experiments will be needed to explore this issue, such as testing ALBCs with higher concentrations of fosfomycin (if it does not impact the mechanical properties of the cement) and investigating the kinetics of the release of this antibiotic, in the presence or absence of gentamicin.

Other interesting approaches have been reported as alternatives to antibiotics for inclusion in PMMA cement. Robu et al. showed that peppermint oil and silver nanoparticles could be incorporated into bone cement, with good biocompatibility and promising antimicrobial effects [31]. Another recent paper by Jackson et al. reported synergistic and extended antibacterial activity by combining gentamicin and silver nitrate in bone cement, which could be of interest for fighting GN-PJIs [32]. Moreover, antimicrobial elution from PMMA cements could be enhanced by varying the composition of the ALBC, notably, by increasing the porosity of the cement as reported by Chen et al. [33]. Bone cements and ALBCs could also be vectors for other drugs that need local delivery to the prosthesis environment. Lüdemann et al. recently reported that adding tranexamic acid, an antifibrinolytic molecule used to reduce peri-operative blood loss, to gentamicin-containing ALBC PALACOS R+G did not alter the activity of gentamicin or the compressive strength of the cement [34].

## 5. Conclusions

To conclude, COPAL G+C seems to be the most interesting ALBC of our tested panel for the prevention of biofilm formation by gentamicin-susceptible Gram-negative bacteria. Further studies will be performed to support and complete our results, especially microscopic fluorescent imaging to provide more insights regarding biofilm viability in the presence of G+C cement. However, gentamicin-resistant strains are still a problem, and further studies are needed to identify an antibiotic/antimicrobial to associate with gentamicin for an efficient dual ALBC against Gram-negative bacteria.

## Figures and Tables

**Figure 1 antibiotics-11-00137-f001:**
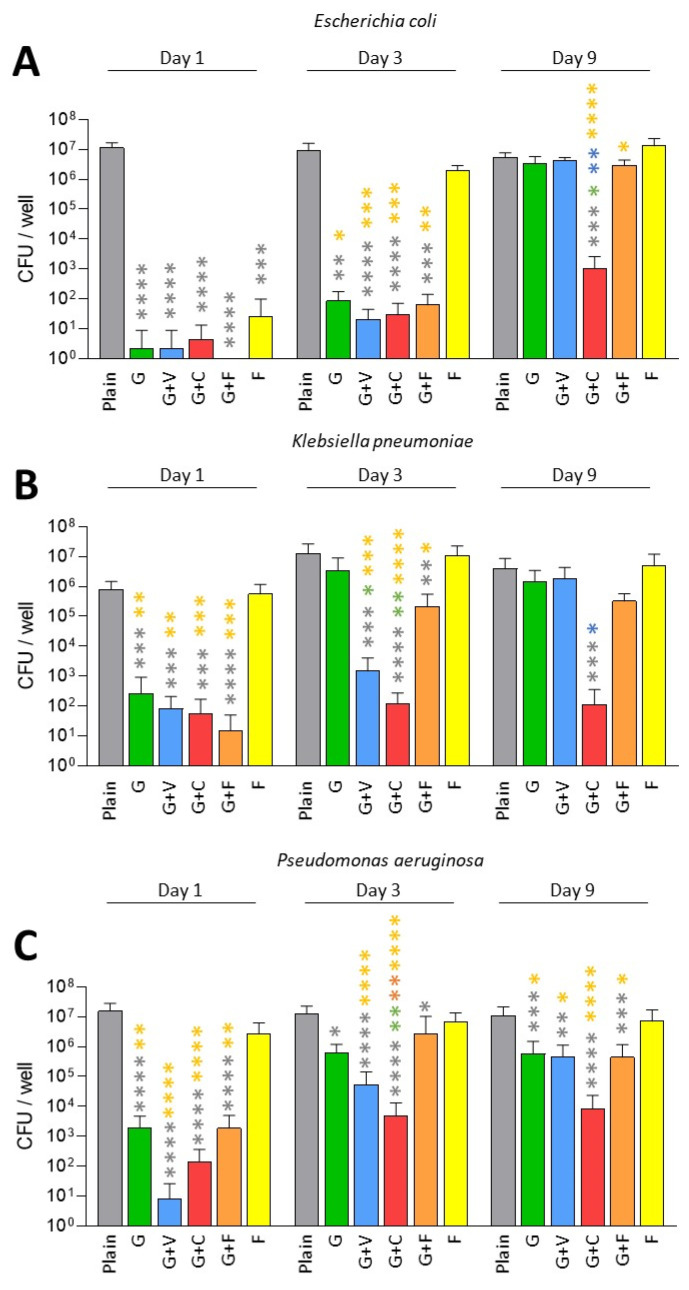
Prophylactic antibiofilm effect of ALBCs against gentamicin-susceptible *E. coli* (**A**), *K. pneumoniae* (**B**) and *P. aeruginosa* (**C**). Results were presented as number of colony-forming units (CFU) in the well after the solubilization of the formed biofilm. Three independent experiments in technical experiments (3 wells for each condition for each experiment) were performed. Non-parametric Kruskal–Wallis tests were performed to compare the data from each day. Dunn’s multiple comparisons tests were performed as follow up tests. For each day, *, **, *** or **** means *p* < 0.05, *p* < 0.01, *p* < 0.001 and *p* < 0.0001, respectively. Color of the stars corresponds to the condition used for the statistical comparison.

**Figure 2 antibiotics-11-00137-f002:**
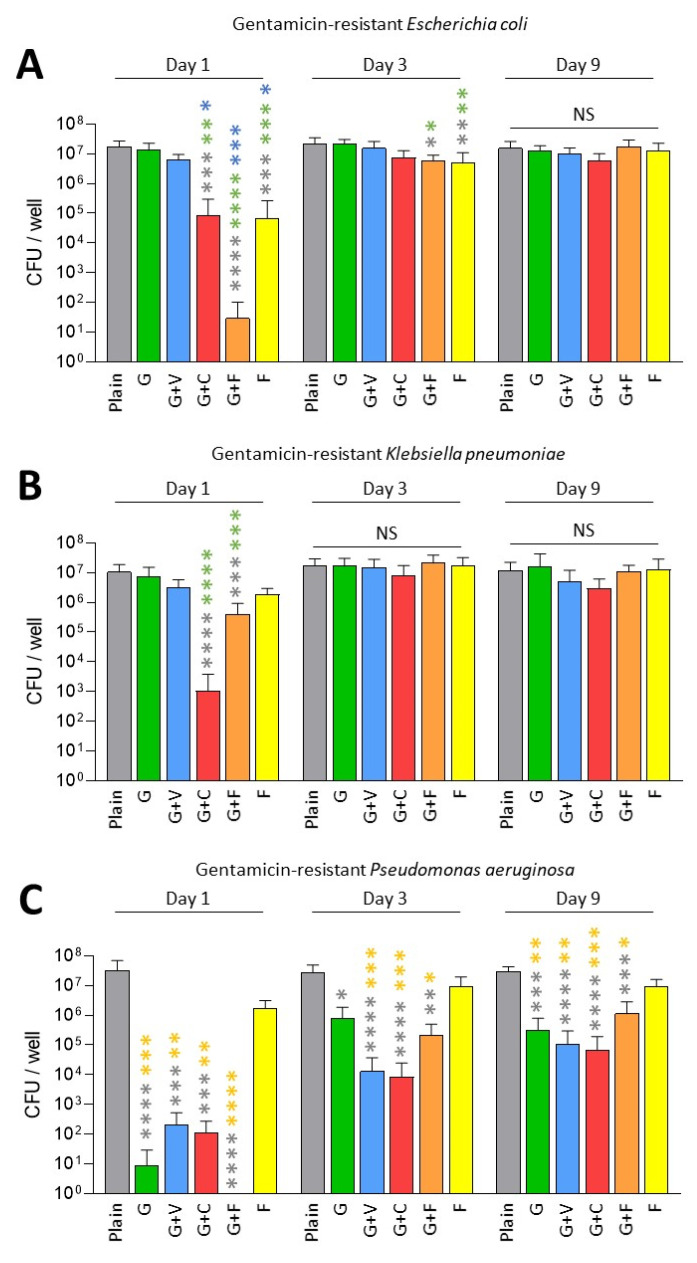
Prophylactic antibiofilm effect of ALBCs against gentamicin-resistant *E. coli* (**A**), *K. pneumoniae* (**B**) and *P. aeruginosa* (**C**). Results were presented as number of colony-forming units (CFU) in the well after the solubilization of the formed biofilm. Three independent experiments in technical experiments (3 wells for each condition for each experiment) were performed. Non-parametric Kruskal–Wallis tests were performed to compare the data from each day. Dunn’s multiple comparisons tests were performed as follow up tests. For each day, *, **, *** or **** means *p* < 0.05, *p* < 0.01, *p* < 0.001 and *p* < 0.0001, respectively. Color of the stars corresponds to the condition used for the statistical comparison. NS means not significant.

**Figure 3 antibiotics-11-00137-f003:**
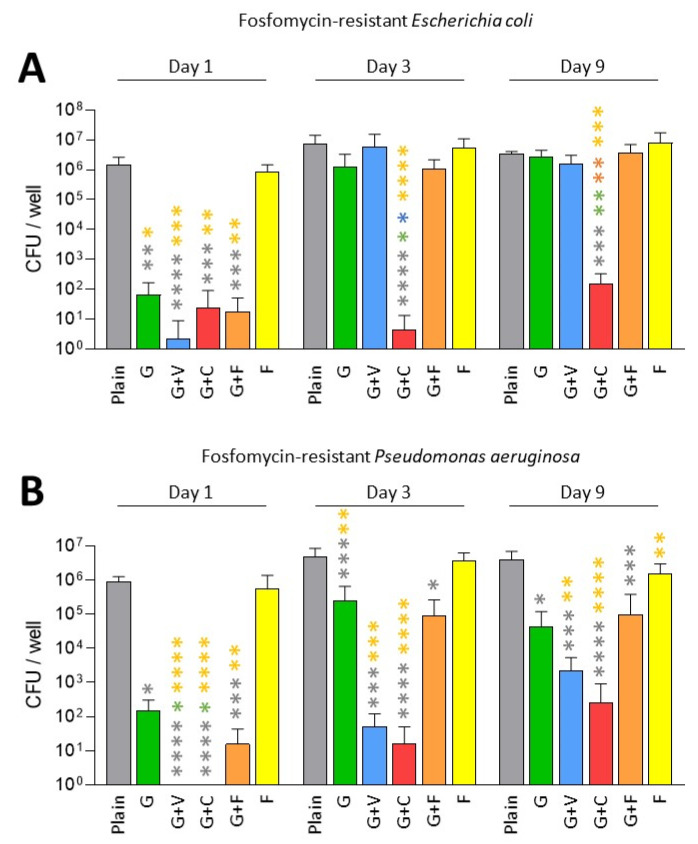
Prophylactic antibiofilm effect of ALBCs against fosfomycin-resistant *E. coli* (**A**) and *P. aeruginosa* (**B**). Results were presented as number of colony-forming units (CFU) in the well after the solubilization of the formed biofilm. Three independent experiments in technical experiments (3 wells for each condition for each experiment) were performed. Non-parametric Kruskal–Wallis tests were performed to compare the data from each day. Dunn’s multiple comparisons tests were performed as follow up tests. For each day, *, **, *** or **** means *p* < 0.05, *p* < 0.01, *p* < 0.001 and *p* < 0.0001, respectively. Color of the stars corresponds to the condition used for the statistical comparison.

**Table 1 antibiotics-11-00137-t001:** Antibiotic resistance of the strains tested in this study.

	Gentamicin	Fosfomycin
*Escherichia coli*	S	S
GentaR *E. coli*	R	S
FosfoR *E. coli*	S	R
*Klebsiella pneumoniae*	S	S
GentaR *K. pneumoniae*	R	S
*Pseudomonas aeruginosa*	S	S
GentaR *P. aeruginosa*	R	S
FosfoR *P. aeruginosa*	S	R

**Table 2 antibiotics-11-00137-t002:** Characteristics of bone cements tested in this study.

Cement	Antibiotic and Quantity				Commercial Name
Plain	-	-	-	-	PALACOS R
G	gentamicin	0.5 g	-	-	PALACOS R+G
G+V	gentamicin	0.5 g	Vancomycin	2 g	COPAL G+V
G+C	gentamicin	1 g	Clindamycin	1 g	COPAL G+C
G+F	gentamicin	0.5 g	Fosfomycin	1.5 g	Not commercialized
F	-	-	Fosfomycin	1.5 g	Not commercialized

## Data Availability

The data presented in this study are available on request from the corresponding author. The data are not publicly available due to public/private collaboration.

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
