# Peer review of "Prophylactic Antibiofilm Activity of Antibiotic-Loaded Bone Cements against Gram-Negative Bacteria"

_antibiotics, 2022, doi:10.3390/antibiotics11020137_

Round 1
Reviewer 1 Report
This is an important topic for the management of biofilms. Though the manuscript is concise, I think it needs further improvement particularly in the methodology and discussion part. The 96-well plate method for biofilm quantification is fine but additional microscopic techniques may provides more insights for bacterial cell viability analysis. The following points need to be taken into consideration for revision
- What method used to identify the biofilm bacterial strains?
- Try add more information in the discussion.
Author Response
Dear reviewer,
Thank you for your comments. We try to improve our manuscript by adding some information in the introduction, adding more details in the results and precise our thoughts in the discussion.
To answer your specific comments (lines will be indicated for “simple marks” revision mode in Word):
- Performing microscopic imaging could be very interesting to provide more insights regarding biofilm viability. However, we currently don’t have cement specimen for these experiments. We hope we can perform these experiments if the manufacturers could provide us adequate specimens. We added sentences in the conclusion regarding this point (lines 316-317)
- Regarding identification of bacterial strains, it was performed at the Hospital by MALDI-TOF. We had some information about that on lines 98-100
- We developed our discussion by adding information about gentamicin + clindamycin cement (lines 276-286) and by adding some examples of alternative antimicrobials that could be use in PMMA cement (lines 303-314).
We hope our revisions will satisfy you.
Reviewer 2 Report
The manuscript entitled “Prophylactic anti-biofilm activity of antibiotic-loaded bone cements against Gram-negative bacteria” presents results on the antibiofilm activity of some bone cements, both commercial and novel samples, loaded with 4 antiobiotics and combinations.
The manuscript is very interesting and provide interesting experimental data, however there are some aspects to be improved and major revision performed before the acceptance. See below the comments:
- Some recent references should be added, since there are a lot of new studies involving for example Gentamicin loaded bine cements.
- Line 62.
“However, adding vancomycin or clindamycin as previously reported will not be effective as these 2 antibiotics have no antibacterial effect on gram-negative bacteria”.
Thus, why the authors aim in the work to investigate mixtures with vancomycin or clindamycin? The authors should clarify this.
- Table 2. The author should provide an explanation for the different amounts of antibiotics used in the bone cement samples in the study.
- Line 127 The concentration of gentamicin in sample G+C is twice compared to other samples with gentamicine. I do not think it is correct to compare the results of samples tested with different concentrations of gentamicin and various added antibiotics. I suggest the authors to try experiments with custom made cement prepared with suitable amount of G+C instead the commercial on, or reconsider the discussion on Fig. 1.
- It is unusual to entitle the different section of the Results as a conclusion statement, e.g.: “Addition of fosfomycin to gentamicin in ALBCs did not provide a higher prophylaxis against biofilm formation by gentamicin- and fosfomycin-susceptible strains”. I suggest to rename the sections with shorter and suitable titles
- Lines 192-200. The paragraph is most relevant in the Introduction section, should be removed from the section Discussions
- In both Discussion and Introduction sections the authors should provide more details on the novelty and relevance of the work presented in the manuscript.
Author Response
Thank you for your comments. We try to improve our manuscript by adding some information in the introduction, adding more details in the results and precise our thoughts in the discussion.
To answer your specific comments (lines will be indicated for “simple marks” revision mode in Word):
- We added some recent examples about gentamicin bone cements, especially examples of alternative antimicrobials that could be use in PMMA cement (lines 303-314).
- In this manuscript, we only focus on ready-to-use ALBC that were directly provided by the manufacturers, even for the G+F and F cement. As we are in a context of prophylactic effect, we don’t know what kind of bacteria could provoke a PJI so the ALBC have to protect against both gram positive and gram negative bacteria. Our question was : does ready-to-use antibiotic protect against gram-negative bacteria and does clindamycin and vancomycin could influence gentamicin effect (especially release) by their presence in the cement. We add some information about this point in the discussion (lines 276-286) and specified that the ALBCs are all ready-to-use and directly provided by the manufacturers
- We hope we answered this point by specifying the ready-to-use character of the tested ALBCs
- As explained above, we rewrote the results and try to be more precise regarding the difference of gentamicin concentration in the G+C cement. As explained in 2, we also discussed this point (lines 276-286)
- Thank you for your remark. We modified it.
- We removed this paragraph from the discussion and reused some sentences in the introduction
- We had some information in introduction and discussion, we hope it will answer your comment.
We hope our revisions will satisfy you.
Reviewer 3 Report
The manuscript describes good results regarding the development of antibiotic-loaded bone cement (ALBC) with different antibiotic concentrations. The authors managed to obtain interesting and positive results with gentamicin and clindamycin-loaded ALBC. The research subject is quite interesting, the study is well planned and executed, although it is preliminary research.
For further improvement, some major corrections are required:
- Please define the abbreviation in the abstract at line 19 – ALBCs
- e. line 57 & 67 or Gram-negative or gram-negative, please use it uniformly and revise the whole manuscript
- Please format references according to journal guideline
- line 96 – please correct to italics et al
- Please put a space between number and ℃ i.e. line 100, line 105…
- line 103 –please define the BHI abbreviation
- line 107 – please correct “a smooth” to “with a smooth”
- line 109 – please correct and define the manufacturer and country of the instrument – Bactosonic
- line 120 - Please define the version and producer of GraphPad software
- line 141 – in both figures the CFU is not defined, and maybe the authors could correct to Number of viable cells (CFU/ml?) – and under the figure the CFU – abbreviation has to be defined
- in the discussion section, the authors should include some additional discussions regarding similar studies.
- in the references correct the microorganism names to italics
Overall, the manuscript presents important experiments and results with significant data which should be described and discussed additionally. After some major improvements and corrections, the manuscript can be published.
Author Response
Dear reviewer,
Thank you for your comments. We try to improve our manuscript by adding some information in the introduction, adding more details in the results and precise our thoughts in the discussion.
To answer your specific comments (lines will be indicated for “simple marks” revision mode in Word):
- We defined ALBC line 19
- We chose Gram negative and kept it all over the manuscript
- Sorry, we submit as free format but we corrected the format references in this version
- We corrected the italic for “et al”
- We added spaces between number and °C all over the manuscript
- We added “with a smooth”, now line 132
- We added manufacturer and country for Bactosonic and all the material
- We specified the version and producer of GraphPad, now lines 144-145
- We corrected the Y axis of all our figures and add information on this point in each caption
- We add some references regarding studies that incorporate alternative antimicrobial in PMMA cement, we hope it will answer your comment
- Microorganisms names have been put in italics in the references
Round 2
Reviewer 2 Report
The manuscript could be published in the revised form provided by the authors.
Reviewer 3 Report
The authors implemented every comment and improvement and considerably improved the manuscript which is now suitable for publication. So considering these aspects after my opinion it can be published.